# Cellulose-Based Hectocycle Nanopolymers: Synthesis, Molecular Docking and Adsorption of Difenoconazole from Aqueous Medium

**DOI:** 10.3390/ijms22116090

**Published:** 2021-06-04

**Authors:** Bayan Khalaf, Othman Hamed, Shehdeh Jodeh, Roland Bol, Ghadir Hanbali, Zaki Safi, Omar Dagdag, Avni Berisha, Subhi Samhan

**Affiliations:** 1Chemistry Department, Faculty of Science, An-Najah National University, Nablus P.O. Box 7, Palestine; bayan.kh107@hotmail.com (B.K.); g.hanbali@najah.edu (G.H.); 2Institute of Bio and Geosciences, Agrosphere (IBG-3), Forschungszentrum Jülich GmbH, 52425 Jülich, Germany; r.bol@fz-juelich.de; 3Chemistry Department, Faculty of Science, Al Azhar University-Gaza, Gaza P.O. Box 1277, Palestine; zaki.safi@gmail.com; 4Laboratory of Industrial Technologies and Services (LITS), Department of Process Engineering, Height School of Technology, Sidi Mohammed Ben Abdallah University, P.O. Box 2427, Fez 30000, Morocco; omar.dagdag@uit.ac.ma; 5Department of Chemistry, Faculty of Natural and Mathematics Science, University of Prishtina, 10000 Prishtina, Kosovo; avni.berisha@uni-pr.edu; 6Materials Science-Nanochemistry Research Group, NanoAlb-Unit of Albanian Nanoscience and Nanotechnology, 1000 Tirana, Albania; 7Research and Development Center, Palestine Water Authority (PWA), Ramallah P.O. Box 2174, Palestine; subhisamhan@yahoo.com

**Keywords:** water treatment, persistent pesticides, difenoconazole, cellulose nanocrystalline, 2,6-pyridine-dicarbonyl dichloride, 2-furan carbonyl chloride

## Abstract

The goal of this work was to develop polymer-based heterocycle for water purification from toxic pesticides such as difenoconazole. The polymer chosen for this purpose was cellulose nanocrystalline (CNC); two cellulose based heterocycles were prepared by crosslinking with 2,6-pyridine dicarbonyl dichloride (Cell-X), and derivatizing with 2-furan carbonyl chloride (Cell-D). The synthesized cellulose-based heterocycles were characterized by SEM, proton NMR, TGA and FT-IR spectroscopy. To optimize adsorption conditions, the effect of various variable such as time, adsorbent dose, pH, temperature, and difenoconazole initial concentration were evaluated. Results showed that, the maximum difenoconazole removal percentage was about 94.7%, and 96.6% for Cell-X and Cell-D, respectively. Kinetic and thermodynamic studies on the adsorption process showed that the adsorption of difenoconazole by the two polymers is a pseudo-second order and follows the Langmuir isotherm model. The obtained values of ∆G ° and ∆H suggest that the adsorption process is spontaneous at room temperature. The results showed that Cell-X could be a promising adsorbent on a commercial scale for difenoconazole. The several adsorption sites present in Cell-X in addition to the semi crown ether structure explains the high efficiency it has for difenoconazole, and could be used for other toxic pesticides. Monte Carlo (MC) and Molecular Dynamic (MD) simulation were performed on a model of Cell-X and difenoconazole, and the results showed strong interaction.

## 1. Introduction

The global concern about pollution caused by persistent pesticides emerges from their ability to bio-accumulate and bio-magnify in the environment, as well as causing many adverse effects for mankind [1,2,3]. The techniques reported in the literature for adsorption of pesticides from water include UV photolysis, electrochemical degradation, adsorption, fenton oxidation, ultrasonication, membrane technology, photo catalyst, and others. Among these treatment techniques, adsorption showed obvious advantages such as reusability, lower cost and ease of operation [3,4,5,6,7,8,9,10,11,12]. Removing of difenoconazole by remediation was also investigated in previous studies. Several adsorbents for difenoconazole extraction from water were also suggested, such as cyclodextrin-based adsorbents, mesoporous activated carbon and graphene-based compounds [13,14,15,16,17].

Difenoconazole has a solubility in water of (15 mg/L) at room temperature. It is a very persistent pesticide with half-life in soil of more than seven months. This synthetic fungicide is used to prevent fungal diseases in plants owing to its systemic and broad-spectrum antifungal activities, for instance, basidiomycetes, ascomycetes and deuteromycetes. It is also an ergosterol biosynthesis inhibitor which is used during the entire growing season of crops [1,18]. Difenoconazole is considered an inhibitor of aromatase activity in human adrenocortical carcinoma cell line H295R. Due to its different molecular configurations, enantiomers of this pesticide probably exhibit stereoselectivity in the process of distribution, toxicity, absorption and degradation in the environment. Previous studies on the fungicidal activity of difenoconazole implied that the biological activities of difenoconazole had enantioselectivity with the (2R,4S)-difenoconazole, with a more pronounced fungicidal activity than other forms [18,19].

Comparing difenoconazole with other triazole fungicides, it possesses relatively high acute toxicity, so the contamination of aquatic ecosystems by difenoconazole is of great environmental concern. In addition, exposure to water polluted by the pesticide difenoconazole has hazardous effects on human health that may cause death in the case of high concentrations [1,18,19,20].

One of the most abundant biological and renewable materials worldwide is cellulose, which can be used as an industrial feedstock for many derivatives and in an unlimited number of applications. Additionally, cellulose with modified surface is considered of great interest as it has many commercial applications [21,22,23,24,25,26,27,28,29]. This study aims to purify water from difenoconazole persistent pesticide using cellulose nanocrystalline cross-linked with 2,6-pyridine dicarbonyl dichloride (Cell-X) and cellulose derivatized with the heterocyclic compound 2-furan carbonyl chloride (Cell-D). The effects of variables such as adsorbent dose, pH, temperature, pesticide concentration and time were studied. In addition, isotherms, adsorption kinetics and thermodynamics for adsorption using both polymer Cell-X and Cell-D were investigated.

## 2. Results and Discussion

### 2.1. Polymer Analysis

Cellulose crosslinking with 2,6-pyridine di-carbonyl dichloride and acylation of cellulose carried in DMAc/LiCl solvent mixture are shown in Figure 1. These chemical reagents were chosen because of their functional groups in which the presence of the aromatic heterocyclic and the carbonyl and chloride groups make them perfect adsorbents for difenoconazole removal from water, such that these functional groups bind difenoconazole pesticide through various physical interactions including dipole–dipole interaction, H-bonding and π–π stacking [30].

#### 2.1.1. FT-IR Analysis

The target cellulosic polymers Cell-X and Cell-D were characterized by FT-IR analysis, and the spectrum obtained for Cell-D is shown in Appendix A. The C=C of the furan ring showed a small band at 1576 cm^−1^. The peak at 1179.74 cm^−1^ corresponds to C-O-C of cellulose (β-glycosidic linkage). The peak at 1717.61 cm^−1^ corresponds to the ester carbonyl, and the low frequency of the carbonyl groups could be related to conjugation with the furan ring. The FT-IR of Cell-X shows almost similar bands except for the disappearance of the ester band and the appearance of the amide band that showed up at 1650 cm^−1^. The similarities between FT-IR spectra for both Cell-X and Cell-D polymers could be due to both having the same polymer back bones and functionalities.

#### 2.1.2. H NMR Spectroscopy

The prepared nano-polymer Cell-D was characterized by ^1^H NMR. The spectrum of Cell-D is shown in Appendix A, and DMSO-d_6_ was used as solvent. ^1^H NMR analysis shows three peaks between 6.71–8.12 ppm, attributed to aromatic furan protons, whereas the cellulose peaks appear between 2.7–5.0 ppm. The four peaks between δ 3.1 and 3.60 are from the four C2, C3, C4 and C5 protons, and the peak at δ 3.1 is consistent with the chemical shift of H_2_. The spectrum also detects downfield peaks at δ 4.35 which contributed to the proton H^1^ at the anomeric carbon. The two multiple peaks at δ 3.92 and 3.85 are assigned to the 2H’s at C-6. These results are consistent with those previously reported [31,32].

We were not able to obtain proton NMR for crosslinked cellulose Cell-X because of a solubility issue which could be contributed to by the crosslinking.

#### 2.1.3. SEM Analysis

SEM images of Cell-D and Cell-X were obtained at three magnifications (1, 10, and 100 µm), shown in Figure 2 and Figure 3.

The obtained SEM images of Cell-X show a porous structure, with high surface. These characteristics contribute to the high adsorbent efficiency of the polymer. In addition, SEM analysis of Cell-D (Figure 3), shows almost similar morphology to that detected for Cell-X.

#### 2.1.4. TGA Analysis

Thermogravimetric analysis was carried out to study the thermal stability of Cell-X and Cell-D. Appendix A show the single step weight loss behavior of both polymers. The degradation temperature of both polymers was observed at around 250 °C. A notable difference between the TGA analysis of the two polymers is that the weight loss from Cell-X was lower at 250 °C, which could be related to cross-linking that adds stability to the polymer.

### 2.2. Adsorption

The effect of variables on adoption efficiency of Cell-X and Cell-D are listed below. The effect of each variable was evaluated while marinating the other variables at optimum-values.

#### 2.2.1. Adsorption Time

The adsorption of difenoconazole from water Cell-X or Cell-D was carried out as a function of time. The other variables such as temperature, pH, adsorbent dose and initial concentration were kept constant at (20 °C, pH 7, 10.0 mg as adsorbent dose and 10.0 ppm as difenoconazole concentration). The results are summarized in Figure 4; as shown, the adsorption started rapidly then leveled off at about 22 min for both polymers Cell-X and Cell-B. During the initial time, the binding sites are available and become occupied as time progresses. The percentage of removal of difenoconazole reached about 87% and 86% for Cell-D and Cell-X, respectively. Acylation of cellulose Cell-D and crosslinked cellulose Cell-X make them unique adsorbents as they are able to trap difenoconazole pesticide through various intermolecular forces including π–π interaction, dipole–dipole and H-bonding.

#### 2.2.2. pH Value

The solution pH is a critical factor in adsorption, since by varying the pH value, the polymer and the adsorbate surface charge will be controlled by the pH effect. In this study the removal efficiency increased along with the pH values as shown in Figure 5, then started to drop at a pH value higher than 8.0. At low pH value, the difenoconazole N are in ammonium form (−NR_3_H^+^), so the adsorption efficiency was the lowest. As the pH reached 6.0, the Ns started to deprotonate and the adsorption efficiency reached the highest pH value of 6.5 for Cell-X and 7.5 for Cell-D. This could be attributed to the availability of a lone pair of electrons on the N, binding strongly with the polymer binding sites. This highest removal was about 88.1% and 88.7% for Cell-X and Cell-D, respectively.

#### 2.2.3. Initial Concentration of Difenoconazole

The initial concentration’s effect on adsorption efficiency was also evaluated, while the other variables (T, t, pH, and adsorbent dose) were kept constant at 20 °C, 10 mg as adsorbent dose, adsorption time: 20 min and 60 min for Cell-D and Cell-X, respectively, pH 8 and 6 for Cell-D and Cell-X, respectively. The highest percentage removal of difenoconazole at an initial concentration of 10.0 ppm was 88.2% using Cell-D and 92.0% using Cell-X (Figure 6).

At low concentration of difenoconazole, the driving force for adsorption was the adsorbate concentration [33], so as the initial concertation increases, the adsorption increases. Then, the limiting factor for adsorption becomes the availability of the active binding sites, controlled by the adsorbent dose. According to the results shown in Figure 6, at 10.0 ppm the binding sites are saturated, and at concentration higher than 10.0 ppm the efficiency tends to drop.

#### 2.2.4. Temperature

The temperature effect on percentage removal of difenoconazole by Cell-X and Cell-D was also evaluated. As above, the other variables were kept constant at optimum conditions (for Cell-D: pH 8, 20 min, 10 ppm difenoconazole concentration and 10 mg as adsorbent dose) and (for Cell-X: pH 6, 60 min, 8 ppm difenoconazole concentration and 10 mg dose). The results are summarized in Figure 7. The maximum adsorption efficiency was determined to be at room temperature; as the temperature increased, the percentage for removal started to decrease. The results indicate a spontaneous exothermic adsorption.

#### 2.2.5. Adsorbent Dose

The dosage that provided the highest adsorption efficiency for difenoconazole was determined in this part of the study. This was accomplished by conducting the experiment on a 10 mL solution of each difenoconazole at a concentration of 10.0 and 8.0 ppm for Cell-D and Cell-X, respectively, a pH value of 8 for Cell-D and pH of 6 for Cell-X, and contact time of 20 min for Cell-D and 60 min for Cell-X, at room temperature. The effect of the adsorbent dosage on the removal of difenoconazole is presented in Figure 8. As shown, the percentage removal of difenoconazole increased along with increase in the Cell-X and Cell-D dosage. The percentage removal reached about 96.5% and 94.2% for Cell-X and Cell-D, respectively. This could be attributed to the type of adsorption mechanisms, which can be described as diffusion and physical interaction. As the dosage increases, the number of binding sites available for interaction increases, and so does the removal efficiency. When all coordination sites on the adsorbent surface are occupied, the diffusion process begins, and in this case could be controlled by an osmosis-like process, the adsorption efficiency leveling off as the equilibrium state is attained [34].

### 2.3. Adsorption Isotherm

Two adsorption isotherm models were evaluated for the current adsorption process Freundlich and Langmuir isotherms [35]. Adsorption kinetic was also conducted using three different models: pseudo-first order, pseudo-second-order and intra-particle-diffusion. A Van’t Hoff graph was plotted to determine the thermodynamic terms ∆H, ∆G and ∆S, whose values give an indication of the thermal dependence of the process (spontaneous at room temperature, or endothermic).

#### 2.3.1. Langmuir Adsorption Model

The Langmuir model represents the chemisorption process. It is assumed that the adsorption sites all have the same amount of energy [36]. This model is presented in Equation (1).
(1)$$\frac{C_{e}}{q_{e}}=\frac{1}{{\mathrm{bQ}}_{o}}+\frac{1}{Q_{o}}C_{e}$$
where C_e_: equilibrium difenoconazole concentration (mg L^−1^), Q_o_: adsorption capacity at equilibrium (mg/g), q_e_: amount of difenoconazole divided by the polymer mass (mg/g), b: Langmuir affinity constant (L/mg), and a graph of (C_e_/q_e_) versus C_e_ results in slope (1/Q_o_) and y-intercept (1/bQ_o_) [35,36].

#### 2.3.2. Freundlich Adsorption Model

This model assumes adsorption on heterogeneous surfaces of or on surfaces with sites of various affinities. The Freundlich isotherm detects that strong binding sites are occupied first, in which the binding strength decreases with the increasing degree of site occupation. This adsorption model is presented in Equation (2) [36].
(2)$${\mathrm{logq}}_{e}={\mathrm{logK}}_{F}+\frac{1}{n}{\mathrm{logC}}_{e}$$
where n is the heterogeneity coefficient, and from its value it can be concluded if an adsorption is favorable or not (g/L); K_F_: Freundlich Isotherm constant (mg/g). Equation (2) represents a straight line equation, so plotting log q_e_ versus log C_e_ results in log K_F_ as an intercept and slope equal to 1/n [36]. Difenoconazole adsorption by Cell-X or Cell-D was fitted to Langmuir and Freundlich isotherms, the results are shown in Figure 9 and Figure 10, and the isotherm parameters are shown in Appendix A. Results detect that difenoconazole adsorption fitted to Langmuir adsorption, since R^2^ of Cell-X and Cell-D were very close to 1.

The values of R^2^ obtained by Langmuir isotherm were close to 1, an indication of the high affinity for Cell-X and Cell- D of difenoconazole, and the adsorption process follows the Langmuir isotherm model.

#### 2.3.3. Adsorption Kinetic Models

Kinetic models can be used to determine the adsorption mechanism of removal of adsorbate by adsorbent, and whether the adsorption process is chemically or physically controlled. The adsorption kinetic models that are evaluated in this work were pseudo-first-order, pseudo-second-order and Intra–Particle–Diffusion-Kinetics [37,38,39].

Pseudo–First-Order–Kinetics

The Pseudo-First-Order equation is presented below
(3)$$\log\left(q_{e}-q_{t}\right)={\mathrm{logq}}_{e}-\frac{K_{1}}{2.303}t$$
where q_e_ and q_t_ are the masses of difenoconazole per adsorbent mass unit at equilibrium at time t, respectively, the units or the terms are (mg/g), and k_1_ is the rate adsorption constant of pseudo-first-order kinetic (mg g^−1^ min^−1^).

Plotting log (q_e_−q_t_) versus time must produce a straight line with log q_e_ as y-intercept and (−k_1_/2.303) as slope [40].

Pseudo-Second-Order–Kinetics

This model shows that the rate-determining step can be fitted to a chemical adsorption process in which the electrons can be shared or exchanged between the pesticide and the adsorbent. The pseudo-second-order-model is presented in Equation (4): (4)$$\frac{t}{q_{t}}=\frac{1}{k_{2}q_{e}^{2}}+\frac{1}{q_{e}}t$$
Where k_2_ is the rate adsorption constant of pseudo-second-order-kinetic at equilibrium with a unit of: g mg^−1^ min^−1^. The graph-produced form plotting t/q_t_ versus t results in a straight line with a slope of 1/q_e_ and y-intercept 1/k_2_q_e_^2^.

Intra-Particle-Diffusion–Kinetics

This model was suggested by Weber and Morris and is shown in Equation (5):(5)$$q_{t}=K_{p}t^{0.5}+C$$
where K_p_ is the rate adsorption constant (mg/g min^1/2^), and C is a constant that depends on the thickness of the boundary layer (mg/g) [41].

A graph of q_t_ versus t^1/2^ represents an intra-particle-diffusion model graph where C is y-intercept and K_p_ is slope [41,42].

The kinetic experimental data for adsorption of difenoconazole pesticide by Cell-X and Cell-D were fitted to intra-particle-diffusion, pseudo-first-order, and pseudo-second-order models to study the mechanism of each adsorption. Results are presented in Figure 11 and Figure 12.

The q_e_ calculated and experimental method are shown in Appendix A, and comparison between these values shows that the adsorption of difenoconazole by Cell-X and Cell-D follows the pseudo-second-order-kinetics and indicates a strong bonding between difenoconazole and adsorbent.

#### 2.3.4. Adsorption Thermodynamics

Thermodynamic study is required to determine if the adsorption process is spontaneous at room temperature or not. The Change in Enthalpy (∆H), Entropy (∆S) and Gibbs free energy (∆G) were calculated [42,43,44,45] using Equations (6)–(8).
∆G = ∆H − T∆S(6)
∆G = −RT × InK_d_(7)
where R has a value 8.314 J mol^−1^ K^−1^ and represents universal gas constant, K_d_: thermodynamic equilibrium constant that equals (q_e_/C_e_) with a unit of mol or L g^−1^.

Results of Equations (6) and (7) are combined in Equation (8).
(8)$${\mathrm{lnK}}_{d}=\frac{\Delta S}{R}-\frac{\Delta H}{\mathrm{RT}}$$

A graph of InK_d_ versus (1/T) (Van’t Hoff plot) was plotted, the slope representing ∆H/R and y-intercept representing ∆S/R [46]. This graph was used for determining the thermodynamic parameters (ΔG°, ΔS and ΔH) for difenoconazole removal by Cell-X and Cell-D. The plots are shown in Figure 13 and the obtained difenoconazole thermodynamic parameters are shown in Appendix A.

The values of ∆H and ∆G° indicate that the adsorption of difenoconazole by Cell-X and Cell-D is spontaneous and exothermic. In addition, negative ∆S indicates the existence of an associative mechanism, in which no significant change will appear in the polymer’s internal structure during adsorption.

### 2.4. Adsorbents Regeneration

The regeneration of adsorbents was carried out by rinsing with a solution of H_2_SO_4_ (0.1 N), distilled water, and ethanol to ensure completed removal of difenoconazole. Results for three adsorption cycles are shown in Figure 14 and show a small drop in removal efficiency. These results are after the first cycle. This is an indication that the Cell-X and Cell-D cellulose based polymers could be promising materials for removing difenoconazole and other persistent pesticides from water to a drinkable degree.

A representative scheme showing the interaction between Cell-X and difenoconazole is seen in Figure 15A. The scheme explains the high affinity of Cell-X to difenoconazole, the main interaction force is H-binding, and other forces contributing to the strong interaction are π–π stacking between the pi-electrons of difenoconazole and the aryl part of 2,6-pyridine di-carbonyl attached to cellulose polymer chain. As mentioned earlier, Cell-X has a semi crown ether structure where cavities are created in the middle (Figure 15A) and difenoconazole passes through the adsorbent pores and becomes trapped in the cavities (inclusion) [47,48].

Cell-D also showed high affinity to difenoconazole, and a representative scheme showing the interaction between Cell-D and difenoconazole is presented in Figure 15B; the main interaction forces between the sorbent and the adsorbate are H-binding and π–π stacking between the pi-electrons of difenoconazole and the furyl part of 2-furan carbonyl, bonded to the cellulose polymer backbone. Cell-X showed an extra major attraction force which is not present in Cell-D.

### 2.5. Monte Carlo (MC) and Molecular Dynamic (MD) Simulation

In Figure 16 the lowest energy configurations for the cellulose surface and the difenoconazole molecule are presented. The assessable decision on the interaction between the difenoconazole molecule Cell-X polymer surface is executed by calculation of the adsorption energies using the following Equation (9):(9)$$E_{ads}=E_{total}-\left[E_{\mathrm{difenoconazole}}+E_{\left(\mathrm{name}\right)\;\mathrm{surface}}\right]$$
where *E_total_* is the total energy of the system as a result of interaction between Cell-X surface and the difenoconazole molecule; $E_{\mathrm{difenoconazole}}$ and $E_{\left(\mathrm{name}\right)\;\mathrm{surface}}\;\mathrm{is}\;$the system’s energy in the absence and presence of difenoconazole.

The lowest energy pose from MC calculations is obtained through a vast number of randomized configurations. The indications for obtaining the lowest energy pose are evident in Figure 17a (the average energy is remained unchanged—indicated by the yellow box in the Figure 17a). The Monte Carlo calculations (Figure 16) are in good agreement with the experimental results, where the difenoconazole strongly adsorbs onto the cellulose surface. The negative value of the adsorption supports the spontaneity of the adsorption process onto this material [47]. The distribution of the adsorption energies is in a range from −0.5–−37.05 kcal/mol, as shown in the Figure 17b. In general, MD is more reliant on monitoring and recording the process’s dynamics. As indicated by the slight temperature drift on the graph in the Figure 17c, the equilibrium configuration is attained. Figure 18 depicts the time evolution of the lowest equilibrium energy structure for the interaction of the difenoconazole molecule onto a surface. The interaction (adsorption) energy during the MD is evaluated at each time elapse and is presented in Figure 19. The flexibility of the Cell-X surface chains during the difenoconazole interaction causes energy fluctuation throughout the MD. As seen from Table 1, the mean value of the adsorption energy of the difenoconazole molecule onto Cell-X surface during the MD is −86.96 kcal/mol.

## 3. Materials and Methods

### 3.1. Chemicals

All chemicals were obtained from Aldrich Chemical Company in Munich, Germany and used without any further purification. They include anhydrous lithium chloride, anhydrous dimethylacetamide, difenoconazole, 2,6-pyridine dicarbonyl dichloride and 2-furan carbonyl chloride.

### 3.2. Methods

The NMR spectra were recorded on a Bruker 600 MHz spectrometer, equipped with a 5 mm broadband CryoProbe Prodigy. The acquisition parameters are: 90° pulse calibrated at 12 μs, 1.3 s acquisition time, 2 s time of relaxation, no spinning, 300 K, and 2048 scans. The proton NMR spectra were obtained at Forschungszentrum Jülich (Juelich, Germany). The FT-IR spectrum was recorded using FT-IR spectrometry, Nicolet iS5, iD3 with ATR (Thermo Scientific, Japan). Thermogravimetric measurements were carried out using the TGA, Q50 V20.10, Build 36 instrument at a heating rate of 10 °C/min under N_2_ gas. The polymer morphologies were investigated using scanning electron microscopy (SEM, Tubney Woods, JEOL 7400F Oxford Instruments Inca, Abingdon, Oxon OX13 5QX, UK).

High Performance Liquid Chromatography (HPLC) was carried out on Waters 1525 Binary HPLC Pump with Photodiode Array Detector PAD (Waters 2998, USA). Data were analyzed using Breeze QS software with the X TERRA C18 column, 5 μm, 250 mm × 4.6 mm, at a flow rate of 1 m/min. The mobile phase was a combination of 36% distilled water and 64% acetonitrile.

### 3.3. Crosslinking of Cellulose with Pyridine 2,6-Dicarbonyl Dichloride (Cell-X)

A cellulose solution was first prepared by the dissolution process reported in the literature [49]. Cellulose was first activated by suspending a 1.2 g (7.4 mmol/anhydro-glucose unit (AGU) of in 100 mL of distilled water. The suspension was mixed for 2 h at ambient temperature. Cellulose was then collected by suction filtration and resuspended in methanol (100 mL) for 1 h to remove water. To ensure the complete removal of water, the procedure was repeated twice, followed by two successive solvent exchange (25 mL each) with DMAc. The first exchange was performed for 1.0 h, while the second exchange was carried out for 24 h. After each solvent exchange, DMAc was removed by suction filtration. The activation stage at this step was completed and the dissolution of activated cellulose was carried out in a two-necked-round bottomed flask fitted with a condenser. To the cellulose in the flask a solution of anhydrous LiCl (6.5 g) was added in a 100 mL anhydrous DMAc. The suspension was stirred until a clear gel was obtained (about 2 h). To the cellulose solution, trimethylamine (0.5 mL) was added dropwise under nitrogen, followed with a solution of 2,6-pyridine di-carbonyl dichloride (2.448 g, 12 mmol) in a 10 mL DMAc anhydrous. The produced solution was heated for 3 h at 70 °C. The produced precipitate was collected by suction filtration and washed several times with distilled water and ethanol, then dried at room temperature for 24 h.

### 3.4. Acylation of Cellulose with Furan-2-Carbonyl Chloride (Cell-D)

The above experiment (2.3) was repeated exactly, except that a solution of furan-2-carbonyl chloride (2.86 g, 22.0 mmol) in 10 mL anhydrous DMAc was used.

### 3.5. Difenoconazole Adsorption Experiments

Batch experiments were done in plastic vials (50 mL) placed in a shaker immersed in a water bath equipped with thermostat. Solution effects such as pH, time, dosage, difenoconazole initial concentration and temperature effect on the polymer’s efficiency were studied. After each run, the sample was collected using a 10 mL plastic syringe, then filtered through a 0.45 µm filter and subjected to HPLC analysis at 229 nm wavelength to calculate the residual pesticide amount after adsorption. All difenoconazole adsorption experiments were performed in triplicate, and the average of the three runs was determined. The difenoconazole adsorbed amount on Cell-X and Cell-D was then determined using Equations (10) and (11), respectively.
(10)$$R\;\left(\%\right)=\frac{C_{0}-{\;C}_{e}}{C_{0}}$$
(11)$$Q_{e}=\frac{C_{0\;}-{\;C}_{e}}{m}V$$
where C_0_ and C_e_: Initial and equilibrium concentrations (ppm) of difenoconazole in solution, respectively, V: solution volume (mL), m: adsorbent dose (mg), and Q_e_: equilibrium adsorption capacity (mg/L).

### 3.6. Monte Carlo (MC) and Molecular Dynamic (MD) Simulation

The interaction between the surface and the difenoconazole molecule was investigated in the vacuum using a large number of randomized Monte Carlo steps (975,956,401 configurations). The MD calculations continued to use the lowest energy pose provided by MC. Prior to the MD stage, the geometry of the simulation boxes was optimized (tolerance for energy convergence of 1 × 10^−5^ kcal/mol; atom-based summation method used for both electrostatic and van der Waals interactions with a cutoff distance of 15.5, a spline width of 1, and a 0.5 buffer) using the Forcite module included in the Biovia software. MD was performed at 25 °C using the constant volume/constant temperature (NVT) canonical ensemble with a simulation time of 500 ps (1 fs time step) [50,51,52,53]. The Berendsen thermostat maintained the T control. Calculations for MC and MD were performed using the Condensed Phase Optimized Molecular Potential II (COMPASSII) forcefield [54,55,56,57,58,59].

## 4. Conclusions

Two cellulosic based heterocyclic nano-polymers were prepared by reacting nanocrystalline cellulose with the crosslinking agent 2,6-pyridine di-carbonyl dichloride (Cell-X) and 2-furan carbonyl chloride (Cell-D). The structures of the synthesized polymers were characterized by SEM, TGA, proton NMR and FT-IR spectroscopy. The prepared polymers Cell-X and Cell-D were evaluated as adsorbents for removal of difenoconazole from water. The adsorption efficiency was studied as a function of pH value, contact time, temperature, adsorbent dose, and initial concentration of difenoconazole. The results were used to determine the optimum adsorption efficiency. The results show that the percentage removal of difenoconazole pesticide form water using Cell-X and Cell-D can be quantitative. The adsorption mechanism was predicted by Langmuir isotherm model. Kinetic data revealed that the adsorption of difenoconazole obeys the pseudo second order. Thermodynamic data showed free energies with negative values, indicating a spontaneous adsorption process of difenoconazole by Cell-X and Cell-D. Theoretical calculation using Monte Carlo (MC) and Molecular Dynamic (MD) simulation models were conducted to confirm the experimental results of strong interaction and spontaneous adsorption between difenoconazole and the adsorbents.

## Figures and Tables

**Figure 1 ijms-22-06090-f001:**
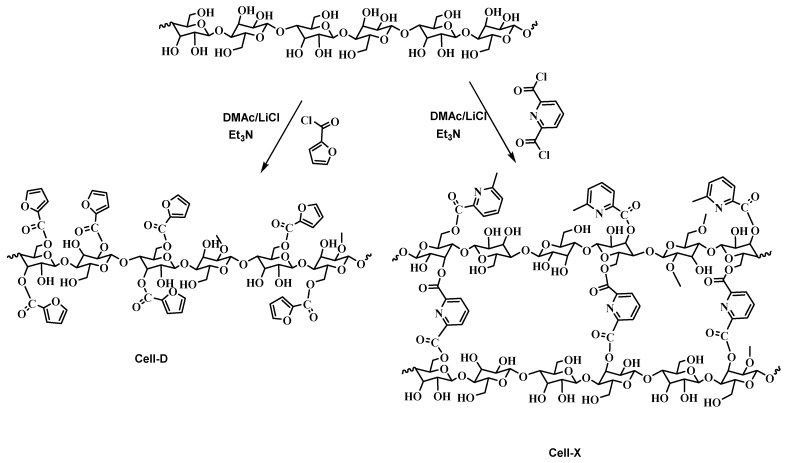
Preparation of Cell-D and Cell-X.

**Figure 2 ijms-22-06090-f002:**
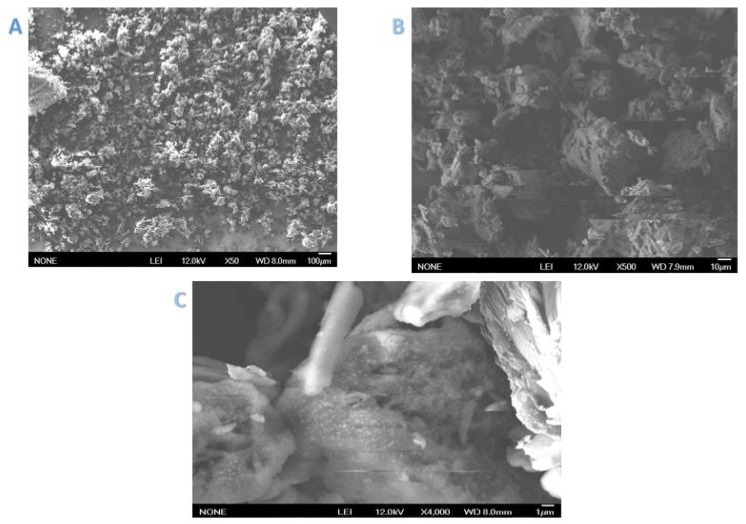
SEM analysis of Cell-X at three different magnifications: (**A**): 100, (**B**): 10, & (**C**): 1 µm.

**Figure 3 ijms-22-06090-f003:**
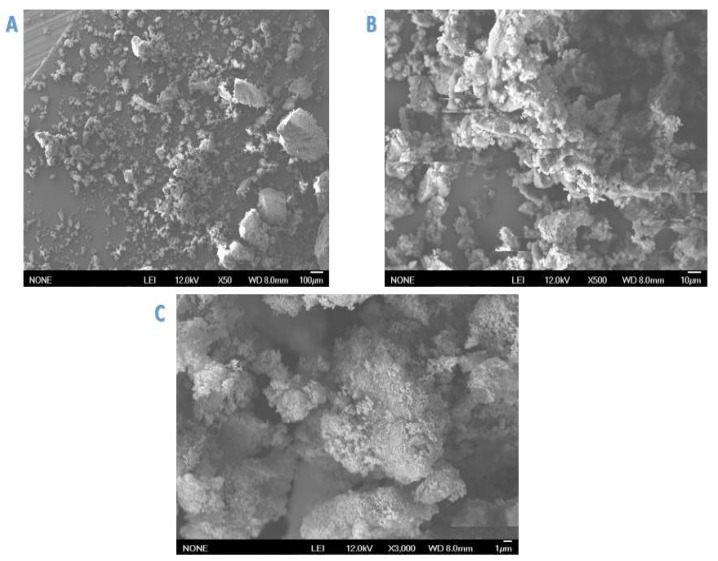
SEM analysis of Cell-D at three different magnifications: (**A**): 100, (**B)**: 10, & (**C**): 1 µm.

**Figure 4 ijms-22-06090-f004:**
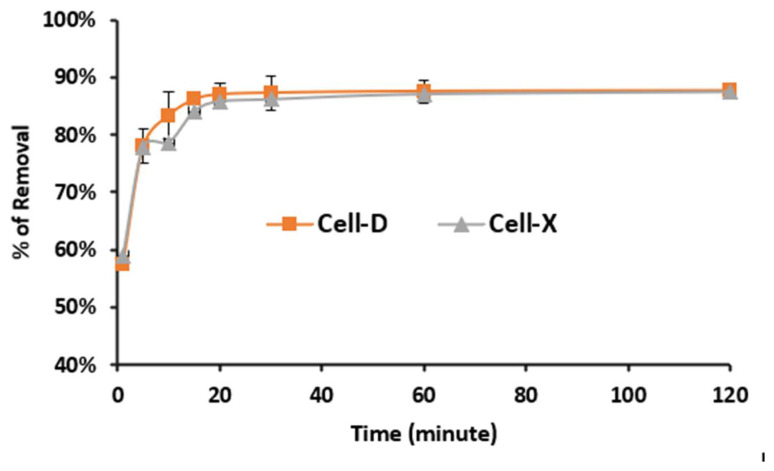
Effect of time on adsorption efficiency of Cell-X and Cell-D (C_o_: 10 ppm, dosage: 10 mg, temperature: 20 °C, volume: 10 mL, pH: 7).

**Figure 5 ijms-22-06090-f005:**
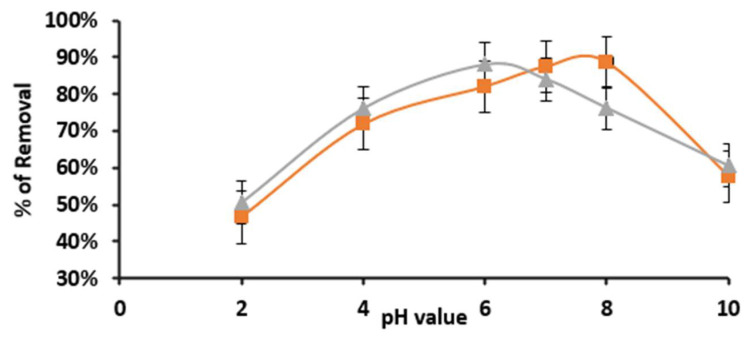
Effect of pH value on adsorption efficiency of Cell-X and Cell-D (C_I_: 10 ppm, dosage: 10 mg, volume: 10 mL, temperature: 20 °C, time: 20 min of Cell-D, time: 60 min of Cell-X).

**Figure 6 ijms-22-06090-f006:**
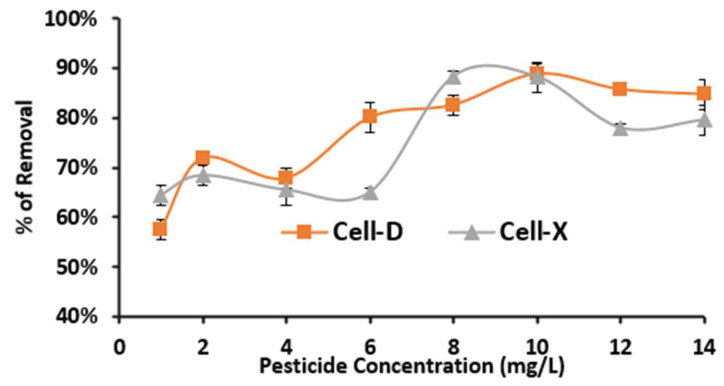
Effect of difenoconazole initial concentration on adsorption efficiency of Cell-X and Cell-D (temperature: 20 °C, dosage: 10 mg, volume: 10 mL).

**Figure 7 ijms-22-06090-f007:**
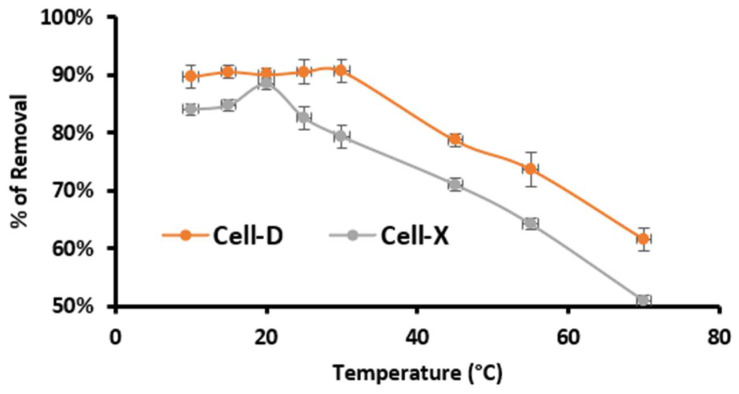
Effect of temperature on adsorption efficiency of Cell-X and Cell-D. (Dosage: 10 mg, volume: 10 mL, pH: 6 and 8 for Cell-X and Cell-D, respectively).

**Figure 8 ijms-22-06090-f008:**
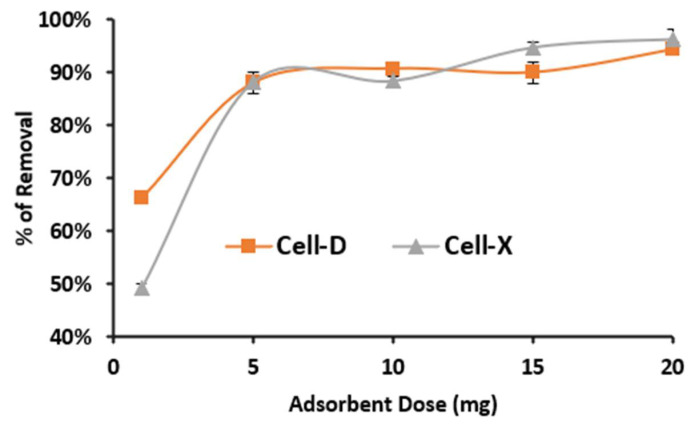
Effect of adsorbent dose on adsorption efficiency of Cell-X and Cell-D. (Volume: 10 mL, 20 °C, pH: 6 and 8 for Cell-X and Cell-D, respectively).

**Figure 9 ijms-22-06090-f009:**
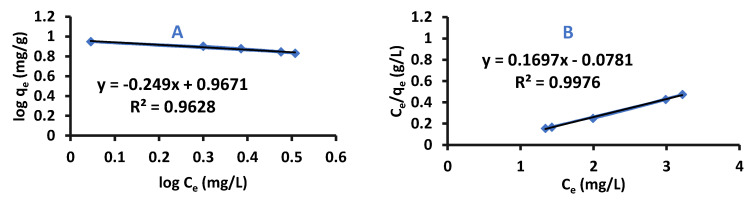
(**A**) Freundlich and (**B**) Langmuir plots for difenoconazole by Cell-D (pH: 4, volume: 10 mL, temperature: 20 °C, adsorption time: 15 min, dosage: 10 mg).

**Figure 10 ijms-22-06090-f010:**
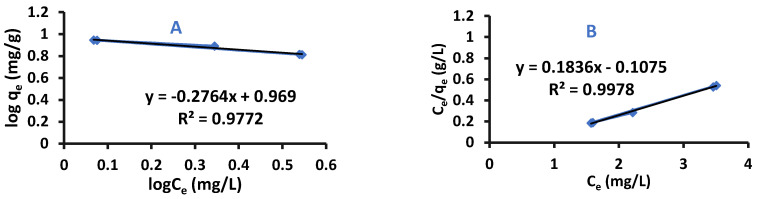
(**A**) Freundlich (**B**) Langmuir plots for difenoconazole by Cell-X (pH: 6, volume: 10 mL, temperature: 20 °C, adsorption time: 20 min, dosage: 10 mg).

**Figure 11 ijms-22-06090-f011:**
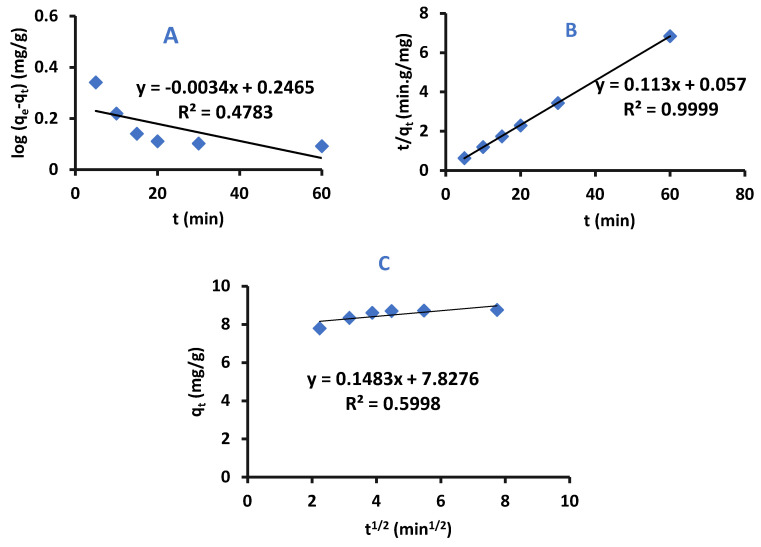
Pseudo-first-order (**A**), Pseudo-second-order (**B**), and Intra-particle-diffusion kinetic (**C**) of difenoconazole removal by Cell-D (C_I_: 10 ppm, volume: 10 mL, temperature: 20 °C, pH: 7, dose: 10 mg).

**Figure 12 ijms-22-06090-f012:**
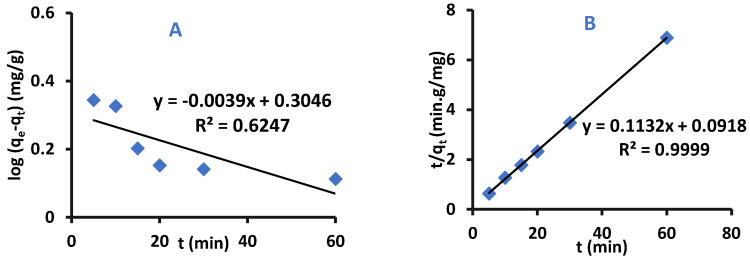
Pseudo-first-order (**A**), Pseudo-second-order (**B**), and Intra-particle-diffusion kinetic (**C**) for difenoconazole removal by Cell-X. (C_I_: 10 ppm, volume: 10 mL, pH: 7, temperature: 20 °C, dose = 10 mg).

**Figure 13 ijms-22-06090-f013:**
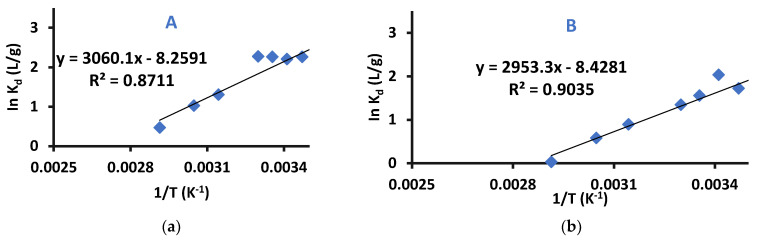
Van’t Hoff plots for the adsorption of difenoconazole by (**A**) Cell-D (C_I_: 10 ppm, pH: 8, time: 20 min), (**B**) Cell-X (C_I_: 8 ppm, pH: 6, time: 60 min), (dosage: 10 mg, volume: 10 mL).

**Figure 14 ijms-22-06090-f014:**
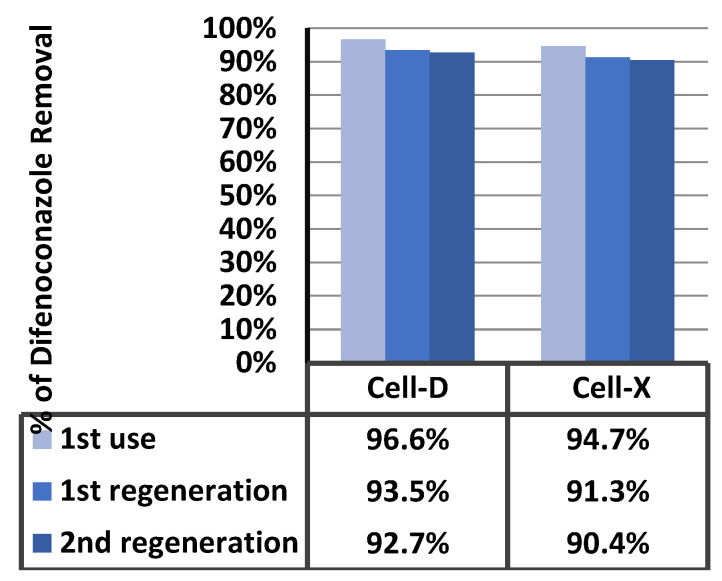
Adsorbent recycling results.

**Figure 15 ijms-22-06090-f015:**
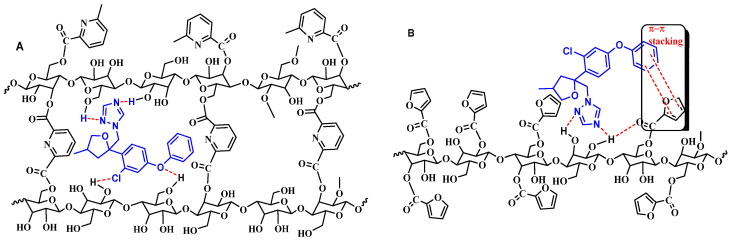
A representative structure shows the interaction between, (**A**) Cell-X and difenoconazole; (**B**) Cell-D and difenoconazole.

**Figure 16 ijms-22-06090-f016:**
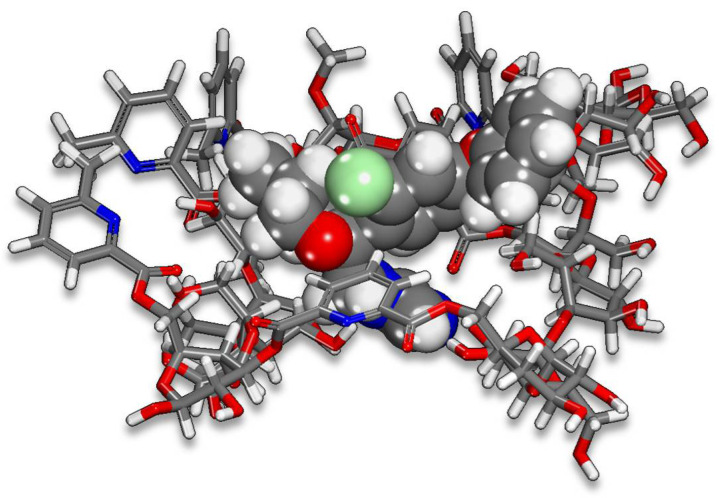
MC pose of the lowest adsorption configurations for the difenoconazole molecule onto the Cell-X surface.

**Figure 17 ijms-22-06090-f017:**
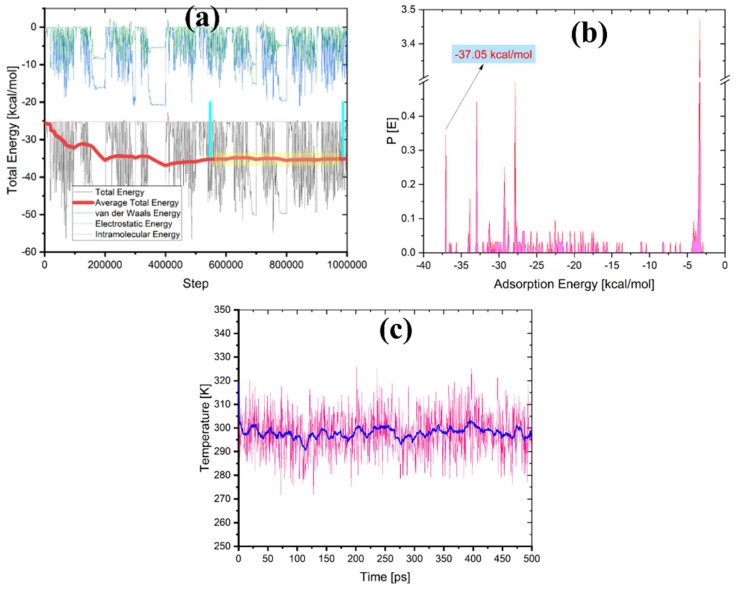
(**a**) Energy terms during the search for the MC pose of the lowest adsorption configurations for the difenoconazole molecule onto Cell-X surface, (**b**) Distribution of the adsorption energies for the difenoconazole molecule onto Cell-X surface gained via MC and (**c**) graph of temperature control from MD during the interaction of the difenoconazole molecule onto Cell-X surface.

**Figure 18 ijms-22-06090-f018:**
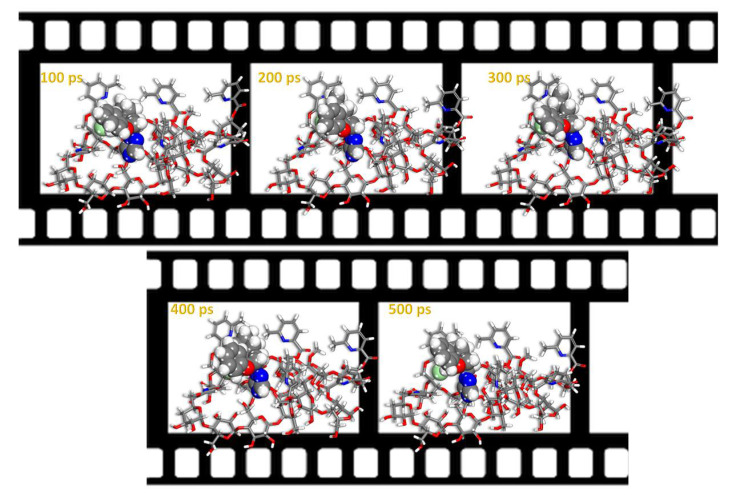
Time-lapse configurations of the interaction of the difenoconazole molecule onto Cell-X surface obtained by MD.

**Figure 19 ijms-22-06090-f019:**
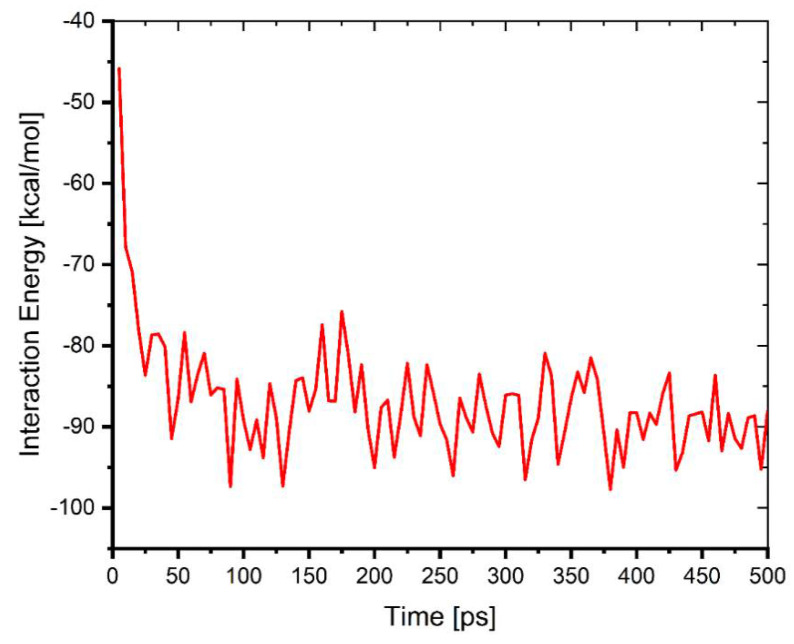
The interaction energy of the difenoconazole molecule onto Cell-X surface during MD. The results show that there are bonding sites on the polymer surface, the difenoconazole molecule fits well and interacts strongly with functional groups present at the bonding sites, and the interaction is strong and stable. The Langmuir Adsorption model also shows that the difenoconazole molecules are equally and homogeneously spread over the polymer surface.

**Table 1 ijms-22-06090-t001:** Statistics for the interaction energy of the difenoconazole molecule onto Cell-X surface during MD.

Mean	Standard Deviation	Minimum	Median	Maximum
−86.9644	6.81479	−97.71934	−88.10748	−45.8472

## Data Availability

Not applicable.

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
