# Peer review of "Cellulose-Based Hectocycle Nanopolymers: Synthesis, Molecular Docking and Adsorption of Difenoconazole from Aqueous Medium"

_ijms, 2021, doi:10.3390/ijms22116090_

Round 1
Reviewer 1 Report
Comments
- Corrections in lines 191 to 193, incomplete sentences.
- Why pH and time duration is different for both the polymers for effect of Difenoconazole initial concentration? When effect of one variable needs to be monitored for two different polymers, ideally all other parameters should be kept constant. It is suggested to provide appropriate rationale and reference for this.
- Same data is repeated in table 1 and figure 9 and 10, similarly for Table 2 and figure 11 and 12 and Table 3 and figure 13. Avoid repetition and reduce length of paper. One can be moved to supplementary file.
- Reference for suggested mechanism of interaction between polymer and Difenoconazole should be added. Mechanism of interaction of Cell D and Difenoconazole and its role in its reduced efficiency should be explored to support the results.
- Conclusion needs improvement. Mere repetition of abstract is not acceptable. Conclusion should not only summarize the studies but should also conclude the work in a clear and crisp manner.
- Line 403 evidenced or evident? Line 405 were or where? Sentence making and grammar needs to be checked.
- Resolution of figure 17 needs improvement.
- Purpose of Monte Carlo (MC) and Molecular Dynamic (MD) simulation is not clear. Explain the meaning of simulations done in terms of specific absorption sites and also explain the meaning of the “mean value of the adsorption energy of the difenoconazole molecule onto (name) surface is -86.96 kca/mol”. How will you correlate the simulations with Langmuir Adsorption model?
- Approximately what will be the shelf life of filters made by this polymer? Considering the effect of various parameters studied in the manuscript, provide an idea of the cost benefits and life or reusability standard of this polymer.
Author Response
- Corrections in lines 191 to 193, incomplete sentences.
The sentence in the lines 191 to 193 has been corrected to be:
A notable difference between the TGA analyses of the two polymers is that the weight loss from Cell-X was lower at 250 oC, which could be related to cross-linking that add stability to the polymer.
- Why pH and time duration is different for both the polymers for effect of Difenoconazole initial concentration? When effect of one variable needs to be monitored for two different polymers, ideally all other parameters should be kept constant. It is suggested to provide appropriate rationale and reference for this.
In the experimental part, we studied the effect of difenoconazole initial concentration for each adsorption process after studying the effect of contact time and pH effect. The values that resulted in the maximum percentages removal of difenoconazole from aqueous solution were as follows: The best pH value was 8 for Cell-D polymer and pH 6 for Cell-X polymer. While the optimum adsorption time was 20 minutes for Cell-D and was 60 minutes. These values were used while studying the effect of the initial concentration.
- Same data is repeated in table 1 and figure 9 and 10, similarly for Table 2 and figure 11 and 12 and Table 3 and figure 13. Avoid repetition and reduce length of paper. One can be moved to supplementary file.
All of the tables (Table 1, Table 2, and Table 3) have been moved to the supplementary file.
- Reference for suggested mechanism of interaction between polymer and Difenoconazole should be added. Mechanism of interaction of Cell D and Difenoconazole and its role in its reduced efficiency should be explored to support the results.
The references for investigating the interaction between the polymers and the pesticide have been added. Such that, the mechanism of this type of interactions includes π-π stacking, dipole-dipole interaction and hydrogen bonding that resulted in having high efficiency in difenoconazole removal from water.
- Conclusion needs improvement. More repetition of abstract is not acceptable. Conclusion should not only summarize the studies but should also conclude the work in a clear and crisp manner.
The conclusion that has been modified.
- Line 403 evidenced or evident? Line 405 were or where? Sentence making and grammar needs to be checked.
The sentences have been corrected.
- Resolution of figure 17 needs improvement.
Regarding, Figure 17 that is the best we can do.
- Purpose of Monte Carlo (MC) and Molecular Dynamic (MD) simulation is not clear. Explain the meaning of simulations done in terms of specific absorption sites and also explain the meaning of the “mean value of the adsorption energy of the difenoconazole molecule onto (name) surface is -86.96 kca/mol”. How will you correlate the simulations with Langmuir Adsorption model?
The reason for carrying out this type of calculation was to show that there is bonding sites on the polymer surface and the difenoconazole molecule fits well and interact strongly with functional groups present in the bonding sites, and the interaction is strong and stable. Langmuir Adsorption model shows the mechanism of the interaction. where the difenoconazole molecule are equally and homogeneously spread over the polymer surface. Some extra explanation has been added to the text.
- Approximately what will be the shelf life of filters made by this polymer? Considering the effect of various parameters studied in the manuscript, provide an idea of the cost benefits and life or reusability standard of this polymer.
The crosslinked polymer Cell-X is expected to have an exceptionally long shelf life. Since crosslinking add stability toughness and strength to the polymer. Also crosslinking adds stability against solvents, mild acidic, and mild basic conditions.

Reviewer 2 Report
The paper titled “Cellulose-Based Hectocycle Nanopolymers: Synthesis, Molecular Docking and Adsorption of Difenoconazole from Aqueous Medium” by B. Khalaf et al. deals with the development of polymer-based heterocycle for water purification from difenoconazole. A multi-analytical characterization of the proposed materials was first accomplished. Then, the adsorption efficiency against difenoconazole was properly evaluated by varying several parameters, such us time, adsorbent dose, pH, temperature, and difenoconazole initial concentration; showing percentage of removal up to 94.7% and 96.6% for Cell-X and Cell-D, respectively.
In my opinion, it is a well-written thorough systematic study that furnishes new insights that can be helpful in the design of versatile high-performance polymer-based systems for water remediation. In addition, it is of high technical quality and includes important information about the thermodynamic parameters describing the adsorption process. Furthermore, the quality of the materials and methods description seems good to me.
Based on the aforementioned considerations the paper deserves, in my opinion, publication in the International Journal of Molecular Sciences - as it positively contributes to the knowledge of the adsorption properties exhibited by such novel cellulose-based nanopolymers.
Some issues are listed below:
- Please, at lines 162-163, add a statement to further specify, according to the synthetization procedure adopted for the production of both Cell-D and Cell-X, why "The FT-IR of Cell-X shows almost similar bands like except the disappearance of the ester band and the appearance of the amide band showed at 1650 cm-1."
- The adsorption of difenoconazole from water Cell-X or Cell-D as a function of time (Figure 4) reveals rapidly increasing trends followed by a plateau. However, in the case of Cell-X, a slight reduction of the removal percentage in the 10-15 min range can be observed. Why, in author opinion, this small reduction occours?
- Please, add ticks in Figure 4 as they seem to be missing both in x and y axis.
- Please, add ticks in Figure 5 as they seem to be missing both in x and y axis.
- Please, a re-phrase of the sentence at lines 231-233 is strongly recommended.
- Please, add ticks in Figure 6 as they seem to be missing both in x and y axis.
- Authors stated that as the number of binding sites available for interaction increases, the removal efficiency increase (lines 264-266). Since both Cell-X and Cell-D reached saturation at the same dosage, it seems like the number of binding sites available for interaction is equal in both Cell-X and Cell-D. Taking into account the chemical structure of each nano-polymer (Cell-X and Cell-D), is this scenario possible?
- Although the writing looks pretty good to me, often an incorrect choice of wording is chosen. Rewriting with a native speaker is recommended.
Author Response
- Please, at lines 162-163, add a statement to further specify, according to the synthetization procedure adopted for the production of both Cell-D and Cell-X, why "The FT-IR of Cell-X shows almost similar bands like except the disappearance of the ester band and the appearance of the amide band showed at 1650 cm-1."
The similarities between FT-IR spectra for both Cell-X and Cell-D nanopolymers could be due to having approximately the same types of functional groups (this sentence has been added to the manuscript).
- The adsorption of difenoconazole from water Cell-X or Cell-D as a function of time (Figure 4) reveals rapidly increasing trends followed by a plateau. However, in the case of Cell-X, a slight reduction of the removal percentage in the 10-15 min range can be observed. Why, in author opinion, this small reduction occurs?
In studying the effect of contact time on difenoconazole adsorption using the nanopolymer Cell-X, the obtained percentages of removal that were detected at 10 minutes and 15 minutes, were 78.63 % and 77.93 %, respectively. This exceedingly small reduction, it could be due to human error and instrument systematic error. However, the obtained results for the triplicate measurements at these two contact times were approximately have the same values of difenoconazole percentages removal.
- Please, add ticks in Figure 4 as they seem to be missing both in x and y axis.
Ticks have been added to Figure 4.
- Please, add ticks in Figure 5 as they seem to be missing both in x and y axis.
Ticks have been added to Figure 5.
- Please, a re-phrase of the sentence at lines 231-233 is strongly recommended.
The sentence has been re-phrased to be: The highest percentages removal for 10.0 ppm difenoconazole initial concentration were 88.2 % in case of using Cell-D and 92.0 % in case of using Cell-X nanopolymer.
- Please, add ticks in Figure 6 as they seem to be missing both in x and y axis.
Ticks have been added to Figure 6.
- Authors stated that as the number of binding sites available for interaction increases, the removal efficiency increase (lines 264-266). Since both Cell-X and Cell-D reached saturation at the same dosage, it seems like the number of binding sites available for interaction is equal in both Cell-X and Cell-D. Taking into account the chemical structure of each nano-polymer (Cell-X and Cell-D), is this scenario possible?
This is an excellent point, we noticed that, and we were supersized too, but this is possible, since we used similar equivalents of cellulose to both reactants and the interaction forces between the two polymers and the adsorbate are also similar.
- Although the writing looks pretty good to me, often an incorrect choice of wording is chosen. Rewriting with a native speaker is recommended.
The manuscript was reviewed, and several corrections and changes have been
Made.

Round 2
Reviewer 1 Report
The authors have addressed the reviewer's comments.